# Shotgun Label-Free Proteomic Analysis for Identification of a Potential Diagnostic Biomarker for Pancreatic Cancer

**DOI:** 10.3390/biomedicines13112631

**Published:** 2025-10-27

**Authors:** Tetsushi Yamamoto, Shogen Boku, Kuniko Mitamura, Atsushi Taga

**Affiliations:** 1Pathological and Biomolecule Analyses Laboratory, Faculty of Pharmacy, Kindai University, Osaka 577-8502, Japan; yamatetsu@phar.kindai.ac.jp (T.Y.); mitamura@phar.kindai.ac.jp (K.M.); 2Cancer Treatment Center, Kansai Medical University Hospital, Osaka 573-1191, Japan; shogen0820@gmail.com; 3Antiaging Center, Kindai University, Osaka 577-8502, Japan

**Keywords:** pancreatic cancer, shotgun proteomics, fibulin-1, serum, diagnosis marker

## Abstract

**Background**: Pancreatic cancer (PC) is one of the cancers with poor prognosis. Although carbohydrate antigen (CA) 19-9 is currently the most specific and sensitive biomarker for the diagnosis of PC and, therefore, the most frequently used, approximately 34% of PC patients may have a Lewis antigen-negative phenotype and secrete little, if any, CA19-9. Therefore, effective alternative diagnostic methods for PC is required. **Methods**: In this study, we examined whether differentially expressed proteins in the blood of PC patients could be identified by global shotgun proteomics analysis using an in vitro cell line sample. **Results**: We identified 142 candidate proteins with differential expression in PC cells. A semiquantitative method and bioinformatic analysis led to a total of 14 candidate proteins that could potentially be detected in blood. Validation studies revealed that the expression of fibulin-1 was lower in PC cells than in normal pancreatic ductal cells. Moreover, in vivo fibulin-1 expression was significantly lower in serum from PC patients than in healthy individuals. **Conclusions**: These findings suggest that lower serum levels of fibulin-1 may be a novel biomarker for the detection of PC.

## 1. Introduction

In 2022, pancreatic cancer (PC) accounted for 511,000 new cancer cases and 467,000 cancer-associated deaths [1]. Despite improvements in treatment methods, the 5-year survival rate for PC patients is still low (<13%) [2,3]; following successful treatment, it is reported to be approximately 20% [4]. Although early detection of PC is important for improving survival rates, it is difficult due to the lack of specific symptoms in its early stages [5]. Therefore, an effective diagnostic method for PC is required.

Currently, carbohydrate antigen (CA) 19-9 is the most specific and sensitive biomarker for diagnosing PC [6,7]. However, patients with a Lewis antigen-negative phenotype, who secrete little, if any, CA19-9, account for approximately 5% to 10% of the population and approximately 34% of patients with PC [8,9,10,11]. Recently, inflammatory factors, such as lymphocytes, monocytes, neutrophils, and C-reactive protein, have been highlighted as potential prognostic markers of PC [12,13,14,15,16]. However, their usefulness as diagnostic markers for this condition, including in early stages of the disease, remains unclear. Therefore, the identification of proteins secreted by PC cells may enable the development of diagnostic methods and improved prognosis.

Proteomics is a useful tool for comprehensively investigating changes in multiple proteins. It has been reported that the vitamin K-dependent protein Z and tumor necrosis factor receptor superfamily member 6b were identified as early diagnostic markers for PC by quantitative proteomic analysis using iTRAQ labeling [17]. Recently, fucosylated SERPINA1 was identified as plasma marker for PC by quantitative glycoproteomic analysis using iTRAQ labeling [18]. In this study, we examined whether proteins whose expression was modified in the blood of PC patients were identified by proteomic analyses using an in vitro normal pancreatic ductal cell line and PC cell lines samples. We identified fibulin-1 as a candidate protein with potential as a novel biomarker for the detection of PC.

## 2. Materials and Methods

### 2.1. Materials

All chemicals and reagents were purchased from Sigma Chemical Corp. (St. Louis, MO, USA).

### 2.2. Cell Culture

The BxPC-3 and PANC-1 PC cell lines and the hTERT-HPNE (HPNE) immortalized normal human pancreatic ductal cell line were purchased from the American Type Culture Collection (ATCC; Manassas, VA, USA). BxPC-3 and PANC-1 cells were cultured in RPMI-1640 medium (Fujifilm Wako Pure Chemical Corporation, Osaka, Japan) supplemented with 10% FBS (Gibco; Thermo Fisher Scientific, Inc., Waltham, CA, USA). HPNE cells were cultured in the medium recommended by the ATCC.

### 2.3. Preparation of Protein Samples for Proteomics and Western Blot

Cells were seeded in a 100 mm dish and were lysed in lysis buffer (7 M urea, 2 M thiourea, 5% CHAPS, and 1% Triton X-100) after 72 h incubation. Culture medium of PC cells was also collected after 72 h. The protein concentration was measured by Bradford assay.

### 2.4. Proteomic Analysis

The proteomic analysis procedure was analogous to that used in a previous study [19,20,21]. The peptides were separated with a reverse-phase C18-column (L-column, 3-μm-diameter gel particles, 120 Å pore size, 0.2 × 150 mm; Chemicals Evaluation and Research Institute). An LTQ ion-trap mass spectrometer (Thermo Fisher Scientific, Inc.) was used for analysis of eluted peptides. Proteomic analysis was performed as three independent experiments for each sample.

The results were searched against the SwissProt Homo sapiens database using Mascot version 2.5.01 (Matrix Science, London, UK). The search criteria were as follows: enzyme: trypsin; allowances: ≤2 missed cleavage peptides; mass tolerance, ±2.0 Da; MS/MS tolerance, ±0.8 Da; cysteine carbamidomethylation; and methionine oxidation modifications. Based on Mascot search results, the score threshold of peptide identification was set to a false discovery rate of 1%.

### 2.5. Semiquantification of Identified Proteins

To compare protein expression across cell line samples, the spectral counting method was used [22]. Fold changes in expressed proteins on a base 2 logarithmic scale were calculated with Rsc [23]. The value of Rsc was determined by utilizing the following equation:Rsc = log_2_(n_p_ + f)/(n_n_ + f) + log_2_(t_n_ − n_n_ + f)/(t_p_ − n_p_ + f)
where n_n_ and n_p_ represent spectral counts for proteins in the normal pancreatic ductal cell line and PC cell line, respectively. t_n_ and t_p_ denote the total numbers of spectra for all proteins in the samples (normal pancreatic ductal cell line and PC cell line), respectively. ƒ is a correction factor set to 1.25.

Proteins with Rsc satisfied >1 or <−1, which corresponded to a fold change of >2 or <0.5, were selected as candidate proteins whose expression is changed in cancer cells.

### 2.6. Bioinformatics

To investigate the functions of differentially expressed proteins with significantly altered expression in PC cells, DAVID (https://davidbioinformatics.nih.gov/, accessed on 28 September 2024) was used for gene ontology (GO) analysis [24,25,26]. A threshold of *p* < 0.05 was considered statistically significant.

The UniProt database (https://www.uniprot.org/, accessed on 30 May 2025) was used to determine which candidate proteins have their subcellular location as a secreted protein.

The UALCAN database (http://ualcan.path.uab.edu, accessed on 28 September 2024) was used to analyze expression of candidates in normal and cancer samples, as well as the effect of fibulin-1 expression on the survival rate of PC patients.

### 2.7. Western Blot

Total protein (10 µg/lane) or culture medium was separated by SDS-PAGE and transferred to PVDF membranes (Millipore Sigma, Burlington, MA, USA) at 15 V for 30 min. The membranes were blocked with TBS + 0.1% Tween-20 buffer (TBS-T) containing 5% skim milk for 2 h at room temperature, and were incubated with the anti-fibronectin (1:1000; cat. no.: ab2413, abcam) antibody, anti-fibulin 1 (1:1,000; cat. no.: ab175204, abcam) antibody, anti-cathepsin D (1:1000; cat. no.: ab6313, abcam) antibody or anti-14-3-3 sigma (1:1000; cat. no.: ab77187, abcam) antibody at 4 °C overnight. After being washed with TBS-T, the membranes were incubated with HRP-conjugated anti-rabbit immunoglobulin G antibody (1:4000; American Qualex, San Clemente, CA, USA) for fibronectin and fibulin-1, HRP-conjugated anti-mouse immunoglobulin G antibody (1:4000; American Qualex) for cathepsin D or HRP-conjugated anti-goat immunoglobulin G antibody (1:4000; Santa Cruz Biotechnology, Inc., Dallas, TX, USA) for 14-3-3 sigma at room temperature for 1 h. The signals were visualized using SuperSignal West Dura Extended Duration substrate (Thermo Fisher Scientific, Inc.) and the ImageQuant RT ECL Imager (Cytiva, Marlborough, MA, USA). The membranes were stripped with Restore Western blot Stripping buffer (Thermo Fisher Scientific, Inc.) and re-probed with anti-β-actin (1:5000; cat. no.: sc-47778, Santa Cruz Biotechnology, Inc.) antibody at 4 °C overnight; this served as the protein loading control. The protein abundance was quantified using densitometry analysis with ImageJ 1.54g software using β-actin as the normalization protein. All Western blot analyses were performed as three independent experiments.

### 2.8. ELISA

Fibulin-1 was quantified in the culture medium and serum by ELISA using the Enzyme-linked Immunosorbent Assay Kit For Fibulin 1 (Cloud-Clone Corp., Katy, TX, USA) according to the manufacturer’s instructions.

### 2.9. Patients and Serum Specimens

Serum samples (n = 10) were collected from patients with PC at Kansai Medical University Hospital between May 2022 and June 2023 with the approval of the Ethics Committee for Kansai Medical University Hospital (IRB approval number 2021279) and Faculty of Pharmacy, Kindai University (IRB approval number 22-197). We obtained informed consent from all patients. Serum samples from healthy donors (n = 6) were purchased from KAC Co., Ltd. (Kyoto, Japan).

### 2.10. Statistical Analysis

Data are presented as the mean ± SEM of at least three independent experiments. Statistical analyses were performed using GraphPad Prism version 8.1.2 (GraphPad Software, Inc., San Diego, CA, USA; Dotmatics). We used unpaired Student’s *t* tests to conduct comparisons between two groups. A *p*-value of less than 0.05 was considered indicative of a significant difference.

## 3. Results

### 3.1. Semiquantitative Comparison of Differentially Expressed Proteins in Normal Pancreatic Ductal Cells and PC Cells

To investigate differentially expressed proteins in PC cells, we performed shotgun proteomics. We identified 724 proteins in the HPNE cells, 761 proteins in the BxPC-3 cells, and 776 proteins in the PANC-1 cells. The overlap of proteins identified among the cell lines is shown in Figure 1.

Next, we used semiquantitative method to identify differentially expressed proteins in the cancer cells. Positive Rsc values indicating increased expression and negative Rsc values indicating decreased expression. A total of 484 and 273 differentially expressed proteins were identified in the BxPC-3 cells and PANC-1 cells, respectively (Appendix A). Among these differentially expressed proteins, the 142 shared altered proteins were identified as PC-related proteins (Appendix A).

### 3.2. Functional Annotation of PC-Related Proteins

We performed a GO analysis of the PC-related proteins to investigate candidate proteins for diagnosing PC. We searched for GO terms related to “Cellular component” (Figure 2), “Molecular function” (Appendix A), “Biological processes” (Appendix A), and “Pathway” (Appendix A) in DAVID, and focused on the 25 proteins classified as “extracellular space” among the Cellular component terms (Appendix A).

Next, we examined the subcellular location of these 25 proteins using the UniProt database to investigate “secreted” protein. Fourteen proteins were assigned to this category (Table 1). Fibronectin and fibulin-1 were downregulated in PC cell lines while cathepsin D and 14-3-3 sigma were upregulated in PC cell lines (Table 1) and were consequently selected as candidate diagnostic biomarkers.

### 3.3. Expression of Candidate Proteins in PC Cells

To validate the results of our proteomic analysis, we examined the expression of candidate proteins in PC cells using Western blot analysis. Concurrent with the proteomics data, the expression of fibronectin and fibulin-1 in the total protein extract from each cell line was markedly lower in the BxPC-3 and PANC-1 cells than in the HPNE cells, and the expression of cathepsin D and 14-3-3 sigma a in the total protein extract from each cell line was markedly higher in the BxPC-3 and PANC-1 cells than in the HPNE cells (Figure 3A).

Then, we examined the candidate proteins secretion level in the culture medium to clarify whether the candidate proteins were secreted into the extracellular space using Western blot. The lower expression of fibronectin and higher expression of 14-3-3 sigma in culture medium from BxPC-3 and PANC-1 cells were observed while there was no clear difference in fibulin-1 expression in the culture medium from each cell line (Figure 3B). On the other hand, no expression of cathepsin D was detected in the culture medium from any of the cell lines tested (Figure 3B). Then, we performed ELISA to confirm that there was no difference in the fibulin-1 secretion level in the culture medium. The fibulin-1 concentration tended to be lower in culture medium from the PC cell lines than in the culture medium from HPNE cells (Figure 3C).

Although decreased expression of fibronectin in PC cells was observed in this study, it has been reported that fibronectin expression in PC tissue was higher than adjacent normal tissues [27]. In addition, it has also been reported that the plasma level of 14-3-3 protein sigma in PC patients was increased as compared to the healthy control [28]. Therefore, we focused on the decreased expression of fibulin-1, whose expression in the blood of PC patients has not been reported, among the other four candidates.

### 3.4. Expression of Fibulin-1 in PC Patients

The serum fibulin-1 concentration in chemotherapy-naïve PC patients was determined by ELISA to investigate whether the protein is suitable as a biomarker for the diagnosis of PC. Detailed information regarding age, TNM classification, and other relevant characteristics of all individuals included in the study can be found in Appendix A. The serum fibulin-1 concentration was significantly lower in PC patients than in healthy individuals (Figure 4, *p* < 0.05).

Next, we examined the fibulin-1 expression in PC patients and its correlation with prognosis using the UALCAN database based on the TCGA database. We did not observe a significant difference in the expression of fibulin-1 between healthy individuals and PC patients (Appendix A; https://ualcan.path.uab.edu/cgi-bin/TCGAExResultNew2.pl?genenam=FBLN1&ctype=PAAD, accessed on 28 September 2024), or a significant correlation between poor prognosis and fibulin-1 expression (Appendix A; https://ualcan.path.uab.edu/cgi-bin/TCGA-survival1.pl?genenam=FBLN1&ctype=PAAD, accessed on 28 September 2024). On the other hand, fibulin-1 expression tended to be low in cancer patients compared to healthy individuals (Appendix A; https://ualcan.path.uab.edu/cgi-bin/Pan-cancer.pl?genenam=FBLN1, accessed on 28 September 2024).

## 4. Discussion

In this study, we identified 142 differentially expressed proteins in PC cells compared to normal pancreatic ductal cells. We also identified 14 candidates that could potentially be detected in blood through bioinformatics analyses based on GO and protein database analysis. We selected 2 upregulated proteins and 2 downregulated proteins as candidate diagnostic markers among these 14 candidates. A validation study revealed that our semiquantitative data obtained from proteomic analysis reflected differences in the expression of each protein among the cell lines. In contrast with previous report [27], fibronectin expression in BxPC-3 and PANC-1 cells was lower than HPNE cells. However, since adjacent normal tissues with enhanced MACC1 expression showed higher fibronectin expression than PC tissues in this report [27], it is possible that MACC1 expression in HPNE cells is also enhanced. In addition, unlike cell lines, tissues are influenced by the surrounding microenvironment. This may be one of the reasons why fibronectin expression differed between cultured cells and tissues reported previously. It is also important to quantify the amount of fibronectin in the serum of PC patients.

On the other hand, verification studies led us to identify fibulin-1 as a biomarker candidate that can be detected through diagnostic blood tests. Consistent with this result, serum fibulin-1 levels were lower in PC patients than healthy individuals. Therefore, we successfully identified candidate protein whose expression level was changed in a clinical sample such as serum via a proteomic approach using an in vitro cell line sample. In addition, it was suggested that decreased fibulin-1 levels in blood may be a useful biomarker for diagnosing PC.

Fibulin-1, a member of the fibulin family [29], is a secreted glycoprotein that interacts with extracellular matrix (ECM) proteins [30] and may be involved in the regulation of signal transduction cascades [31]. In the present study, we found that fibulin-1 expression was significantly lower in PC patients than in healthy individuals although only a limited number of serum samples were available. Several reports have already investigated the expression of fibulin-1 in blood and its usefulness as a biomarker. Plasma fibulin-1 expression has been reported to be significantly lower in papillary thyroid cancer patients than in healthy individuals [32]. In addition, the serum fibulin-1 level has been found to be significantly lower in colorectal cancer patients than in patients with benign colon polyps or healthy individuals [33]. These findings suggest that fibulin-1 secretion from cell might be suppressed through oncogenesis. On the other hand, the TCGA database analysis showed no significant difference in fibulin-1 expression between PC patients. However, the TCGA dataset of PC patients used in this study had a low number of normal cases (n = 4) despite including 176 cancer cases. In addition, TCGA database analysis focused on other types of cancers showed that the expression of fibulin-1 in cancer patients tended to be decreased. Therefore, an increase in the number of normal cases in the TCGA dataset for PC increases might provide confirmation of decreased fibulin-1 expression. It is important to conduct further studies to clarify the specificity and/or sensitivity of fibulin-1 as a diagnostic marker for PC by examining its expression in serum in a larger number of PC cases and comparing its effectiveness with existing PC marker.

The tumor-suppressing effect of fibulin-1 has been reported for many types of cancers, including gastric, prostate, and estrogen-dependent cancers [34,35,36,37,38,39,40]. However, several studies have reported that fibulin-1 has oncogenic functions [41,42,43]. Recently, Aksoy et al. reported fibulin-1 plays a role in PC based on immunohistochemical analysis [44]. No significant difference in fibulin-1 expression was observed between PC patients and pancreatitis patients, which is consistent with our results based on the TCGA database. On the other hand, fibrin-1 expression in the cancer microenvironment of PC was associated with prognosis, and the survival rate was significantly lower in the high fibulin-1 expression group than in the low fibulin-1 expression group. In addition, recent study reported that fibulin-1 expression was correlated with fibrosis [45]. Since PC is characterized histopathologically by prominent fibrosis in the tumor microenvironment [46], decreased expression of fibulin-1 might be useful marker not only for detecting PC but also for reflecting the degree of fibrosis in PC. It is necessary to investigate the relationship between each stage and the expression of fibulin-1.IFurthermore, alternative splicing of fibulin-1 mRNA results in four different variants (fibulin-1 A to D) that differ in the C-terminal region [47]. Thus, it may be possible to enhance the specificity of PC diagnosis by focusing on the fibulin-1 variants expressed in the blood. Further studies are needed to develop an analytical method that can quantify variants of fibulin-1.

## 5. Conclusions

In conclusion, we successfully identified candidate biomarkers for PC that could potentially be detected in the blood based on a label-free semiquantitative method and GO analysis. A decreased expression of fibulin-1 in the serum of PC patients compared to healthy individuals was observed in validation studies. Therefore, the serum level of fibulin-1 may be a useful detection biomarker for PC.

## Figures and Tables

**Figure 1 biomedicines-13-02631-f001:**
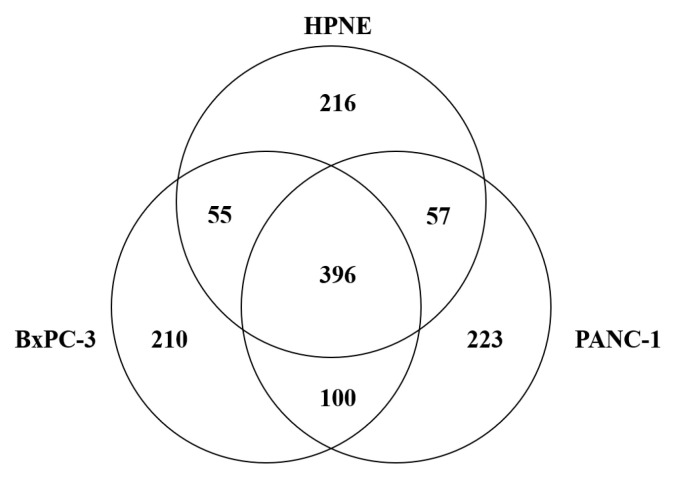
Venn diagram of proteins identified from the normal pancreatic cell line and pancreatic cancer cell lines. A total of 724 proteins were identified in HPNE normal pancreatic cells and 761 and 776 were identified in the BxPC-3 and PANC-1 pancreatic cancer cell lines, respectively.

**Figure 2 biomedicines-13-02631-f002:**
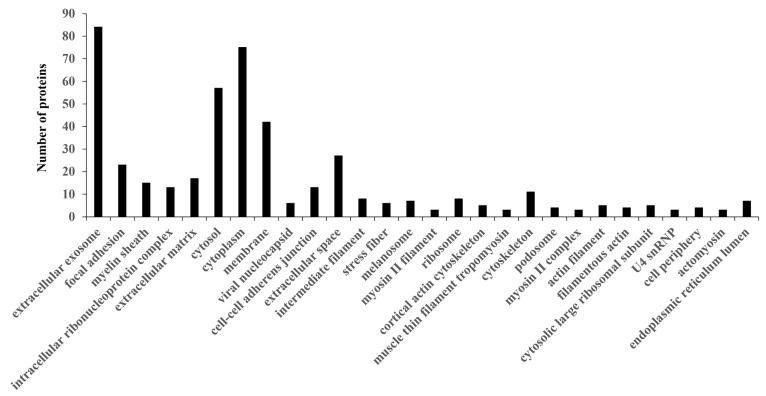
Analysis of proteins identified in the GO cellular component. Proteins assigned to the cellular component category in GO with *p* < 0.05.

**Figure 3 biomedicines-13-02631-f003:**
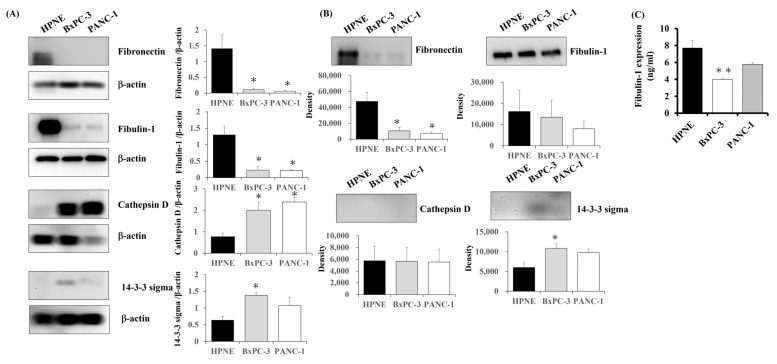
Candidate proteins expression in PC cell lines. (**A**) Western blot of the expression of candidate proteins in cell extracts were similar to the results of proteomic analysis. (**B**) Western blot of the expression of candidate proteins in the culture medium showed secretion of all the candidate proteins except for cathepsin D. Decreased expression of fibronectin and increased expression of 14-3-3 sigma in culture medium of pancreatic cancer cell lines were observed, while the expression of fibulin-1 was not clearly changed. (**C**) Examining the amount of fibulin-1 secreted into the extracellular space by ELISA, the concentration of fibulin-1 was significantly lower in culture medium from BxPC-3 cells than culture medium from HPNE cells and tended to be lower in PANC-1 cells compared to HPNE cells. * *p* < 0.05 ** *p* < 0.01.

**Figure 4 biomedicines-13-02631-f004:**
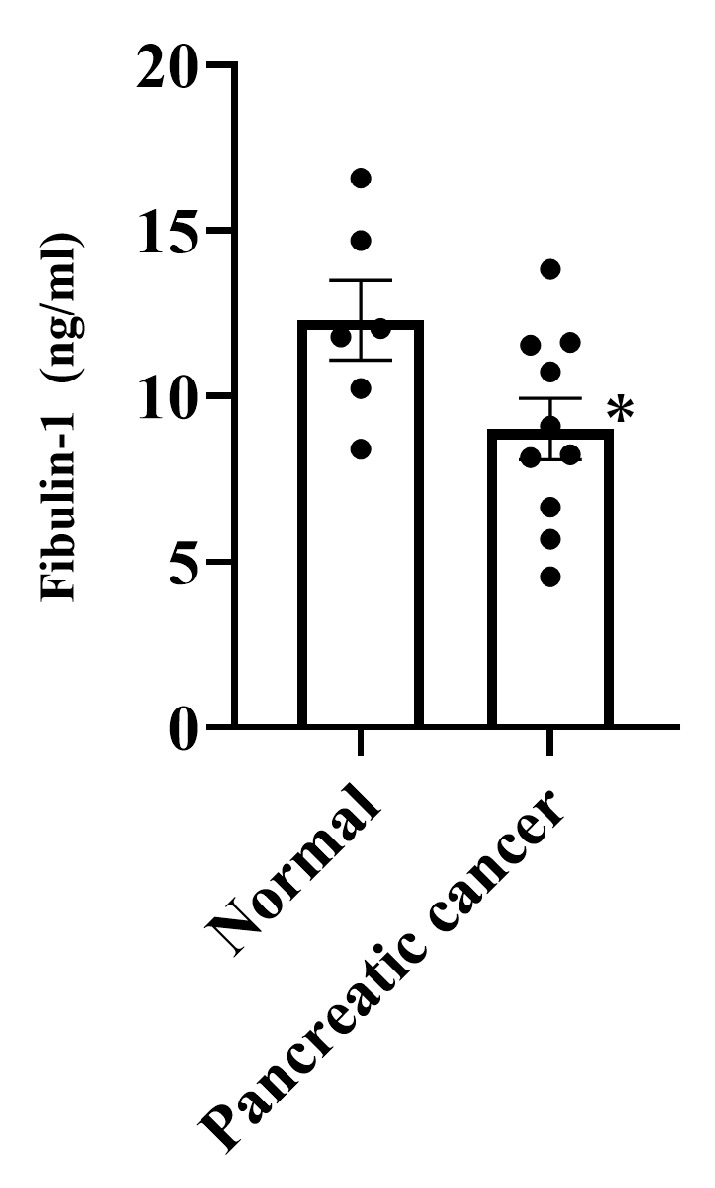
Expression of fibulin-1 in serum from pancreatic cancer patients. Fibulin-1 expression was examined by ELISA of serum from healthy individuals (Normal; n = 6) and pancreatic cancer patients (Pancreatic cancer; n = 10). * *p* < 0.05.

**Table 1 biomedicines-13-02631-t001:** Proteins categorized as secreted proteins.

No.	Protein Name	Spectral Counting			Fold Change (Rsc)	
		HPNE	BxPC-3	PANC-1	BxpC-3	PANC-1
1	Fibronectin	109	1	0	−5.814	−6.532
2	Plasminogen activator inhibitor 1	20	0	0	−4.267	−4.136
3	Collagen alpha-1(I) chain	10	0	0	−3.347	−3.216
4	Fibulin-1	3	0	0	−1.941	−1.810
5	Serpin I2	3	0	0	−1.941	−1.810
6	Collagen alpha-3(VI) chain	3	0	0	−1.941	−1.810
7	Gelsolin	9	2	2	−1.833	−1.702
8	Collagen alpha-1(III) chain	2	0	0	−1.554	−1.422
9	Dermcidin	12	4	3	−1.512	−1.686
10	Calreticulin	14	6	4	−1.249	−1.584
11	High mobility group protein B1	2	7	6	1.170	1.115
12	Cathepsin D	0	3	8	1.592	2.846
13	Glucose-6-phosphate isomerase	2	18	9	2.395	1.615
14	14-3-3 protein sigma	0	33	3	4.608	1.723

Fourteen proteins were categorized according to their subcellular location as secreted proteins via UniProt database analysis.

## Data Availability

The raw MS data files were deposited in the ProteomeXchange Consortium via the jPOST partner repository (http://jpostdb.org, accessed on 28 February 2025) under the dataset identifier PXD061294/JPST003631. Publicly available datasets utilized in this study, such as the TCGA dataset, can be accessed through their respective repositories following the guidelines provided by the data-sharing platforms (fibulin-1 expression in pancreatic cancer; https://ualcan.path.uab.edu/cgi-bin/TCGAExResultNew2.pl?genenam=FBLN1&ctype=PAAD, accessed on 28 September 2024, survival curve; https://ualcan.path.uab.edu/cgi-bin/TCGA-survival1.pl?genenam=FBLN1&ctype=PAAD, accessed on 28 September 2024, fibulin-1 expression in pan-cancer; https://ualcan.path.uab.edu/cgi-bin/Pan-cancer.pl?genenam=FBLN1, accessed on 28 September 2024).

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
