# Peer review of "Shotgun Label-Free Proteomic Analysis for Identification of a Potential Diagnostic Biomarker for Pancreatic Cancer"

_biomedicines, 2025, doi:10.3390/biomedicines13112631_

Round 1

Reviewer 1 Report

Comments and Suggestions for Authors

This manuscript presents a proteomic approach to identify secreted proteins from pancreatic cancer (PC) cell lines, highlighting fibulin-1 as a potential diagnostic biomarker. The work is timely and addresses the urgent need for improved biomarkers in PC. The methodology is sound, and the study integrates proteomics, bioinformatics, and limited clinical validation. However, the sample size is small, the interpretation of findings is sometimes overstated, and certain inconsistencies with public datasets are not fully resolved. With revision and clarification, this study could provide a valuable contribution.

  • The novelty of fibulin-1 as a PC biomarker should be more clearly emphasized. Previous reports have linked fibulin-1 to other cancers; therefore, the unique significance of its role in PC requires stronger justification.
  • The validation cohort (10 PC patients versus 6 controls) is too small to support firm conclusions. This limitation should be explicitly acknowledged in both the Results and Discussion. Details of the clinical cohort (stage distribution, treatment history, comorbidities such as pancreatitis) are essential, as these may affect serum protein levels and biomarker specificity.
  • Serum ELISA results indicate decreased fibulin-1 in PC patients, but TCGA/UALCAN analysis shows no significant difference. This discrepancy requires deeper discussion, including possible explanations such as sample size, population differences, or tumor microenvironmental effects.
  • The discussion should place fibulin-1 in the context of existing PC biomarkers (CA19-9, SPan-1, DUPAN-II, etc.). Could fibulin-1 complement rather than replace these markers? Direct comparison would strengthen clinical relevance.
  • In proteomics, more detail on replicates, normalization in spectral counting, and reproducibility is needed. Also in the western blot/ELISA part, state the number of independent experiments, loading controls, and statistical treatment.
  • The discussion of fibronectin and other proteins should expand on why cell line results differ from tissue data. Differences in tumor microenvironment versus cell culture should be acknowledged.
  • Besides above comments, improve grammar and clarity throughout.

Author Response

Point 1: The novelty of fibulin-1 as a PC biomarker should be more clearly emphasized. Previous reports have linked fibulin-1 to other cancers; therefore, the unique significance of its role in PC requires stronger justification.

Response 1: Thank you for your helpful comment. In response to the reviewer’s comment, we have changed the configuration of the “Discussion” section and have considered the unique significance of fibulin-1 in PC.

Point 2: The validation cohort (10 PC patients versus 6 controls) is too small to support firm conclusions. This limitation should be explicitly acknowledged in both the Results and Discussion.

Response 2: According to the reviewer’s comment, we have added this limitation to our experimental results in the “Results” and “Discussion” sections.

Point 3:  Details of the clinical cohort (stage distribution, treatment history, comorbidities such as pancreatitis) are essential, as these may affect serum protein levels and biomarker specificity.

Response 3: Thank you for your helpful comment. In response to the reviewer’s comment, we have added additional clinical information in “Table S5”.

Point 4: Serum ELISA results indicate decreased fibulin-1 in PC patients, but TCGA/UALCAN analysis shows no significant difference. This discrepancy requires deeper discussion, including possible explanations such as sample size, population differences, or tumor microenvironmental effects.

Response 4: Thank you for your helpful comment. In response to the reviewer’s comment, we have added a discussion on the differences between our results and TCGA analysis.

Point 5: The discussion should place fibulin-1 in the context of existing PC biomarkers (CA19-9, SPan-1, DUPAN-II, etc.). Could fibulin-1 complement rather than replace these markers? Direct comparison would strengthen clinical relevance.

Response 5: Thank you for your helpful comment. As you pointed out, it is important to examine the specificity/sensitivity of fibulin-1 in PC in comparison with existing PC markers. In this study, the use of serum from a limited number of PC patients has been approved as the material obtained from humans. To clarify whether the decreased expression of fibulin-1 is a more useful marker compared with existing markers, it will be necessary to perform a direct comparison in future studies after obtaining new approval. In response to the reviewer’s comment, we have added this information to the “Discussion” section.

Point 6: In proteomics, more detail on replicates, normalization in spectral counting, and reproducibility is needed. Also in the western blot/ELISA part, state the number of independent experiments, loading controls, and statistical treatment.

Response 6: Thank you for pointing this out. In response to the reviewer’s comment, we have added this information to the “Material and methods” and “Results” sections and to the “Figure legend”.

Point 7: The discussion of fibronectin and other proteins should expand on why cell line results differ from tissue data. Differences in tumor microenvironment versus cell culture should be acknowledged.

Response 7: Thank you for your helpful comment. In response to the reviewer’s comment, we have added a discussion of the difference in cell line results and tissue data to the “Discussion” section.

Reviewer 2 Report

Comments and Suggestions for Authors

The authors performed global shotgun proteomics analysis of pancreatic cell lines and identified 142 differentially expressed proteins compared to normal cell lines using spectral counting. Among these, 25 were annotated as extracellular by GO analysis, and 14 were further categorized as secreted proteins according to UniProt. Western blotting confirmed altered expression of several proteins in cell lysates. ELISA further showed that fibulin-1 was decreased in the culture medium of pancreatic cancer cell lines compared to HPNE. Its serum level was also lower in pancreatic cancer patients than in healthy donors, suggesting potential as a detection biomarker. The proteomics workflow is adequately described but outdated, and several major concerns remain:

  1. The overall proteome coverage is very limited.
  2. No correction for multiple testing was applied to the reported p-values of quantitative proteomics.
  3. The definition of “Relative abundance” in Figure 2 is unclear, and the meaning of corresponding p-values are not provided.
  4. Detailed patient and donor information is missing, and the sample size is too small to support strong conclusions.
  5. Results from previously published quantitative proteomics studies in pancreatic cancer are not introduced in the Introduction, nor are they compared with the findings of this study.
  6. Without providing the FDR cutoff, readers cannot judge how reliable the reported 142 differentially expressed proteins actually are.

Author Response

Comment 1: No correction for multiple testing was applied to the reported p-values of quantitative proteomics.

Response 1: Thank you for pointing this out. In this study, differentially expressed proteins were identified by calculating semiquantitative relative changes based on spectral counts. We have added detail on the method used to calculate the relative change values in this study to the “Materials and methods” section.

Comment 2: The definition of “Relative abundance” in Figure 2 is unclear, and the meaning of corresponding p-values are not provided.

Response 2: Thank you for pointing this out. According to the reviewer’s comment, we have changed “Relative abundance” to “Number of proteins” in Figure 2. In addition, we have the meaning of the p-values to the “Materials and methods” section.

Comment 3: Detailed patient and donor information is missing, and the sample size is too small to support strong conclusions.

Response 3: Thank you for pointing this out.  According to the reviewer’s comment, we have added additional clinical information in “Table S5” and have discussed this limitation of our experimental results in the “Results” and “Discussion” section.

Comment 4: Results from previously published quantitative proteomics studies in pancreatic cancer are not introduced in the Introduction, nor are they compared with the findings of this study.

Response 4: Thank you for pointing this out. According to the reviewer’s comment, we have added previously published quantitative proteomics studies on pancreatic cancer to the “Introduction” section.

Comment 5: Without providing the FDR cutoff, readers cannot judge how reliable the reported 142 differentially expressed proteins actually are.

Response 5: Thank you for pointing this out. According to the reviewer’s comment, we have added information on the FDR cutoff in the “Material and methods” section.

Reviewer 3 Report

Comments and Suggestions for Authors

In the manuscript, the authors examined the proteome of blood from PC patients and different cell lines in order to identify potentially diagnostic biomarkers for PC. Proteomic analysis revealed that 14 candidate proteins could be potential diagnostic biomarkers for PC. Future validation experiment using blood of PC patients identified fibulin-1 as a candidate biomarker for the detection of PC. The initial work is interesting, but the experiments conducted are not very clear. Authors should either perform additional experiments, correct the manuscript and explain any inconsistencies regarding the experimental approach

Major

In Figure 1 authors presented Venn diagrams of reported proteins of the normal pancreatic cell line and pancreatic cancer cell lines. In the text authors wrote, “A total of 484 and 273 differentially expressed proteins were identified in the BxPC-3 cells and PANC-1 cells, respectively”. It is not clear how these numbers were obtained. Are unique proteins from cancer cell lines and up-regulated proteins in cancerous cell lines (when compared to control cell line, HPNE) taken into account? It would be logical to first analyze 100 proteins unique to both cancer cell cultures but absent in control cell line. The analysis could then be extended to the 396 proteins to find upregulated proteins in cancer cell lines.

It is not clear differences between “extracellular space” and “secreted” protein. The protein of extracellular exosome should also be interesting to analyze (Figure 2).

Table 2 does not contain unique proteins from cancerous cell lines. Could the authors explain why?

If we analyze the results in Figure 3, it would be logical to check the expression of fibronectin instead of fibulin-1 in serum from pancreatic cancer patients regardless of previous investigations.

Minor

It is not clear why ELISA was performed for the analysis of fibulin-1 if the expression of this protein was checked with Western blot analysis.

The exact p value should be presented in Figure 4. Looking at the values of expression of fibulin-1 ​​in Figure 4, it is not very evident that this protein would be a good biomarker for PC.

The authors could perhaps have used a better performing MS instrument instead of an LTQ ion-trap mass spectrometer. In principle, they should have obtained a significantly higher number of proteins from cell lines.

Comments on the Quality of English Language

NA

Author Response

Comment 1: In Figure 1 authors presented Venn diagrams of reported proteins of the normal pancreatic cell line and pancreatic cancer cell lines. In the text authors wrote, “A total of 484 and 273 differentially expressed proteins were identified in the BxPC-3 cells and PANC-1 cells, respectively”. It is not clear how these numbers were obtained. Are unique proteins from cancer cell lines and up-regulated proteins in cancerous cell lines (when compared to control cell line, HPNE) taken into account? It would be logical to first analyze 100 proteins unique to both cancer cell cultures but absent in control cell line. The analysis could then be extended to the 396 proteins to find upregulated proteins in cancer cell lines.

Response 1: Thank you very much for your helpful comments. In this study, differentially expressed proteins were identified by calculating semi-quantitative values ​​based on spectral counts. In response to the reviewer’s comment, we have added information about the method used to calculate the relative change values in this study to the “Materials and methods” section. On the other hand, although we considered it important to focus on proteins whose expression is up-regulated in cancer cells as you pointed out, we also considered that proteins whose expression is down-regulated in cancer cells are also important as marker candidates. Therefore, we attempted to identify candidates by comparative analysis of protein expression levels.

Comment 2: It is not clear differences between “extracellular space” and “secreted” protein. The protein of extracellular exosome should also be interesting to analyze (Figure 2).

Response 2: Thank you for pointing this out. The “extracellular space” contains secreted protein and proteins, with a section that extends into the extracellular space, such as membrane proteins. Therefore, we performed additional analysis to identify proteins secreted into extracellular space. We also focused on the proteins categorized as the “extracellular exosome” as a useful candidate for diagnostic biomarkers in a future study.

Comment 3: Table 2 does not contain unique proteins from cancerous cell lines. Could the authors explain why?

Response 3: Thank you for pointing this out. Since we indicate only semi-quantitative values in Table 1, it may appear that unique proteins are not included. However, this is not true. In response to the reviewer’s comment, we have added spectral counting values in Table 1.

Comment 4: If we analyze the results in Figure 3, it would be logical to check the expression of fibronectin instead of fibulin-1 in serum from pancreatic cancer patients regardless of previous investigations.

Response 4: Thank you very much for pointing this out. Actually, it is important to quantify fibronectin in the serum of pancreatic cancer patients. In response to the reviewer’s comment, we have added this information to the “Discussion” section. In addition, we have added a discussion on the difference in cell line results and tissue data in the “Discussion” section.

Comment 5: It is not clear why ELISA was performed for the analysis of fibulin-1 if the expression of this protein was checked with Western blot analysis.

Response 5: Thank you very much for pointing this out. The Western blot results suggest that there was no significant difference in fibulin-1 secretion between normal pancreatic ductal cells and pancreatic cancer cells. To confirm whether the secreted fibulin-1 level was no different, we performed ELISA, which is a more quantitative method.

In response to the reviewer’s comment, we have amended the discussion on why we performed ELISA in the “Result” section.

Comment 6: The exact p value should be presented in Figure 4. Looking at the values of expression of fibulin-1 ​​in Figure 4, it is not very evident that this protein would be a good biomarker for PC.

Response 6: Thank you very much for pointing this out. In response to the reviewer’s comment, we have added the p-value.

Round 2

Reviewer 2 Report

Comments and Suggestions for Authors

One of the major concerns mentioned previously is the quantitative proteomics technique used in this study is outdated. In addition, the overall proteome coverage is very limited. However, these major concerns have not been addressed in the current revision.

Reviewer 3 Report

Comments and Suggestions for Authors

It is my pleasure to inform you that your manuscript has been accepted for publication.